# The Effects on Immune Function and Digestive Health of Consuming the Skin and Flesh of Zespri^®^ SunGold Kiwifruit (*Actinidia Chinensis* var. *Chinensis* ‘Zesy002’) in Healthy and IBS-Constipated Individuals

**DOI:** 10.3390/nu12051453

**Published:** 2020-05-18

**Authors:** Sarah L. Eady, Alison J. Wallace, Duncan I. Hedderley, Kerry L. Bentley-Hewitt, Christine A. Butts

**Affiliations:** 1The New Zealand Institute for Plant and Food Research Limited, Private Bag 4704, Lincoln 7608, New Zealand; sarah.eady@plantandfood.co.nz; 2The New Zealand Institute for Plant and Food Research Limited, Private Bag 11600, Palmerston North 4442, New Zealand; duncan.hedderley@plantandfood.co.nz (D.I.H.); kerry.bentley-hewitt@plantandfood.co.nz (K.L.B.-H.); chrissie.butts@plantandfood.co.nz (C.A.B.)

**Keywords:** kiwifruit, immune health, gut health, immune markers

## Abstract

Irritable bowel syndrome (IBS) is a common gastrointestinal disorder that results in constipation (IBS-C) or diarrhoea with abdominal pain, flatulence, nausea and bloating. Kiwifruit (*Actinidia* spp.) are nutrient-dense fruit with a number of reported health benefits that include lowering glycaemic response, improving cardiovascular and inflammatory biomarkers, and enhancing gut comfort and laxation. This study investigated the effect of consuming three whole Zespri^®^ SunGold kiwifruit (*Actinidia chinensis* var. *chinensis* ‘Zesy002’) with or without skin on cytokine production and immune and gut health in healthy people and those with IBS-C symptoms. This study enrolled thirty-eight participants in a 16 week randomized cross-over study (19 healthy and 19 participants with IBS-C). Participants were randomized to consume either three kiwifruit without eating the skin or three kiwifruit including the skin for 4 weeks each, with a 4 week washout in between each intervention. There was a significant decrease in the pro-inflammatory cytokine, TNF-α, for both the healthy and the IBS-C participants when they consumed whole kiwifruit and skin, and also for the healthy participants when they ate whole kiwifruit without the skin (*p* < 0.001). The kiwifruit interventions increased bowel frequency and significantly reduced the gastrointestinal symptom rating scale constipation and Birmingham IBS pain scores for both participant groups. We have demonstrated that consuming the skin of SunGold kiwifruit might have beneficial effects on gastrointestinal health that are not produced by consuming the flesh alone.

## 1. Introduction

Kiwifruit are recognised as a natural food that enhances laxation [1]. Laxation or defecation is an important process in maintaining human health and the feeling of well-being. Laxation induced by dietary fibre results in fermentation occurring more distally in the colon, providing protection against colorectal cancer and other disorders. Irritable bowel syndrome (IBS) is a widespread functional disorder that affects approximately 11% of the world’s population with a significant impact on their quality of life [2]. The presentation and aetiology of the condition are heterogeneous, with individuals presenting with either constipation or diarrhoea and a host of other gastrointestinal symptoms such as abdominal pain, flatulence, nausea and bloating. However, the mechanisms underlying the pathophysiology of the disorder remain unclear [2,3].

The proposed mechanisms include disturbed neural function along the gut–brain axis, abnormalities of motility, visceral sensation, and alterations in gut fermentation. More recently, important roles for low-grade inflammation and immunological changes have also become evident, including mucosal immune activation with the infiltration of mast cells and T cells into the intestinal mucosa and systemic immune activation leading to cytokine imbalance. This is based largely on the observation that there is a higher prevalence of IBS in individuals with prior exposure to an acute enteric infection than in individuals who have not been exposed and a higher prevalence of IBS-like symptoms in patients who have irritable bowel disease in remission [4,5,6,7].

Studies have suggested that cytokines may act as potential aetiological factors in IBS. There are increased plasma or serum concentrations of the cytokines TNF-α, IL-6, IL-8 and lower concentrations of IL-10 in individuals with IBS as opposed to healthy controls [7,8,9]. This has been supported by studies demonstrating polymorphisms in the genes for TNF-α, Il-6, IL-10 and other cytokines, leading to disturbances in the levels of expression of these compounds [10]. The use of probiotics and prebiotics as modulators of the gut microbiome in IBS is becoming widely accepted, with some evidence of efficacy, and there are a number of studies supporting their beneficial effects including inhibiting visceral hypersensitivity, improving intestinal barrier function and reducing symptoms such as flatulence, abdominal distension and gastrointestinal discomfort [11,12]. It is unclear whether probiotics or prebiotics exert their beneficial effects by modulating immune function [4].

Recent research has investigated the role of kiwifruit in gut health in both healthy individuals and those suffering from IBS complicated with constipation. Green kiwifruit (*Actinidia chinensis* var. *deliciosa* ‘Hayward’) [13] and gold Zespri^®^ SunGold kiwifruit (*Actinidia chinensis* var. *chinensis* ‘Zesy002’) [14] were shown to provide relief from constipation and some gastrointestinal symptoms such as bloating. The mechanisms through which they provide these effects are unknown but it is thought that components in kiwifruit, namely non-digestible soluble and insoluble fibre and kiwifruit polyphenols, improve stool frequency and general gastrointestinal health as well as providing functional benefits to gut health through prebiotic and gentle prokinetic effects. These studies used only the flesh of the fruit, as these fruit are usually consumed with the skin removed. The skin is, however, completely edible and may help increase fibre intake and preserve the high vitamin C content of the fruit. The skin contains a number of bioactive substances including tocopherols, sterols, triterpene, chlorogenic acid and flavonoids that have demonstrated some anti-bacterial activity and anti-oxidant properties [15,16].

In the present study, our primary hypothesis was that consuming whole fruit (including the skin) of SunGold kiwifruit would reduce the production and release into the blood stream of the inflammation biomarker C-reactive protein and the cytokines Il-6, IL-10 and TNF-α in individuals with IBS-C, and that this would be different when kiwifruit flesh alone was consumed. Our secondary hypothesis was that whole SunGold kiwifruit (flesh + skin) would reduce the gastrointestinal symptom scores (constipation, diarrhoea, flatulence, and abdominal pain) associated with IBS and this would be greater than for eating only the kiwifruit flesh.

## 2. Materials and Methods

This study was conducted according to guidelines laid down in the Declaration of Helsinki and was approved by the New Zealand Human Disability and Ethics Committee (18STH63). The trial was registered with the Australia New Zealand Clinical Trials Registry (ACTRN: 12618000340235p) and all participants gave written informed consent. The design of this clinical study was a randomized cross-over trial of 16 weeks duration. The interventions were SunGold kiwifruit with and without skin. This kiwifruit has smoother and thinner skin, and so may be more acceptable to consumers than that of other kiwifruit varieties.

The primary outcome for this study was a decrease in the concentration of pro-inflammatory cytokines (TNF-α, IL-6) and an increase in the anti-inflammatory cytokine (IL-10) compared with the baseline period (baseline). The secondary outcomes were improvements in the number of complete spontaneous bowel movements (CSBMs), complete bowel motions (CBMs), spontaneous bowel motions (SBMs), bowel movement frequency (BM), number of strained bowel motions and stool form as well as gut symptoms and comfort using the GSRS and Birmingham IBS score compared with the baseline. A CSBM is when the bowel is emptied completely—that is, the person feels like no more stool needs to come out and the stool came out without having to take laxatives or use manual manoeuvres. A CBM is when the bowel is emptied completely—that is, the person feels like no more stool needs to come out. A SBM is when the person did not have to use laxatives or manual manoeuvres to get the stool out. A BM is the number of times the person passed a stool per week. A strained BM is whether the person had to push really hard to get the stool out. The Bristol Stool Scale (BSS) rates the stool from 1 (very hard, like pellets) to 7 (diarrhoea, watery, mushy stools).

Forty adults were recruited to this study: 20 participants classified as having IBS-C (according to Rome III criteria) and 20 healthy participants. They were recruited through newspaper and radio advertisements, local district health boards and tertiary institution newsletters, existing participant databases and community advertisements. Participants were aged between 18 and 65 years and had a BMI of ≤35 kg/m^2^. Fasting blood glucose concentration was required to be below 6.0 mmol/L.

Healthy participants had normal bowel habits and waist circumference (≤88 cm for European, Māori, and Pacific Island women; ≤108 cm for European, Māori, and Pacific Island men; ≤80 cm for Asian and Indian women; ≤90 cm for Asian and Indian men). IBS-C participants were required to meet the criteria for IBS-C as described by the Rome III criteria [17]. The IBS-C diagnostic criteria for IBS was recurrent abdominal pain or discomfort at least 3 days per month in the last 3 months associated with 2 or more of the following: improvement with defecation, onset associated with a change in frequency of stool; and onset associated with a change in form (appearance of stool). IBS-C requires meeting the IBS criteria together with hard or lumpy stools (Bristol Stool Form Scale 1–2) ≥ 25%, and loose or mushy stools ≤ 25% of bowel movements (Bristol Stool Form Scale 6–7). (Table 1) Participants were excluded if they had any alarming features associated with bowel habit (recent changes in bowel habit (<3 months), rectal bleeding, weight loss, occult blood in stools, anaemia); anal fissures; bleeding haemorrhoids; a family history of GI cancer or inflammatory bowel disease (IBD); chronic disease (cardiovascular, cancer, renal failure, previous gastrointestinal surgery (not including appendectomy or cholecystectomy)); neurological conditions (e.g., multiple sclerosis, spinal cord injury, stroke); blood glucose ≤ 6.1 mmol/L; women who were pregnant, breastfeeding or planning a pregnancy in the 3 months post selection (trial period); known to have kiwifruit or latex allergy; using laxatives and who were not prepared to stop for this study. Participants with diagnosed and stable conditions requiring the use of selective serotonin reuptake inhibitors (SSRIs), tricyclates, opiates or anti-inflammatories were permitted into the trial on the condition that the medication had been in use continually and the condition had been stable for >3 months. Similarly, those with stable and controlled diabetes (>3 months) were permitted to participate.

Participants continued their habitual diet and lifestyle but excluded kiwifruit, high-fibre supplements and laxatives (except the prescribed rescue laxative if needed) for at least 2 weeks before starting this study and during the trial period. Following this 2 week washout, participants were randomized to one of the treatment groups: group 1 consumed three whole (skin and flesh) SunGold kiwifruit daily for 4 weeks and group 2 consumed three SunGold kiwifruit (skins removed) daily for 4 weeks. Participants were allowed to consume the fruit in any way they wished. Baseline measurements recorded for each participant were fasting blood glucose, C-reactive protein, kidney and liver function, acid/base balance, anthropometry (height, weight, BMI), and a dietary eating habits questionnaire.

After the 4 week intervention period, participants entered a 4 week washout phase, in which no kiwifruit were consumed. The participants then swapped to the opposite treatment for 4 weeks before the final washout phase of 2 weeks, in which no intervention was consumed. At the beginning and end of each phase, participants were asked to complete two validated questionnaires: the gastrointestinal symptom rating scale (GSRS; AstraZeneca Ltd., MöIndal, Sweden), which rated their gastrointestinal comfort, and the Birmingham IBS score [18]. They provided a blood sample for cytokine analysis (TNF-α, IL-6, IL-10, C-reactive protein). Throughout this study, the participants completed a daily bowel habit diary that measured their stool frequency and stool consistency, as well as a 3 day diet record at the end of each intervention phase to assess energy and nutrient intake (Copies of these questionnaires are included in the Appendix A). A blood sample was collected for the measurement of glycated haemoglobin (HbA1c) at the start and end of this study.

During the baseline and washout phases of this trial, patients suffering from constipation were able to use their regular treatments to relieve symptoms up until 1 week (7 days) before any intervention phase. Any treatments used in this period were required to be reported in the daily bowel habit diary. In the week (7 days) immediately before intervention treatments, participants were to refrain from the use of any form of treatment to prevent or relieve constipation other than use of the Bisacodyl suppositories, named as the rescue treatment for this trial. In circumstances where functionally constipated participants had no bowel movements (BM) for 3 days, or up to a maximum of 2 BM per week, they were instructed to use Bisacodyl suppositories (5 g) as the rescue laxative. The use of all rescue treatments was recorded in the daily diaries.

The undiluted plasma samples were assayed with Human Inflammation Panel mix and match subpanels from BioLegend, San Diego, CA following the manufacturer’s instructions for analytes TNF-α, IL-6 and IL-10. Samples were analysed on a BD FACSVerse and data were processed using LEGENDplex™ Data Analysis Software (version 8).

A power calculation was undertaken to determine the number of participants that would be required to measure a statistically significant difference. To detect changes in the level of pro-inflammatory cytokines with 80% power and 5% significance and an increase in CSBMs of 1.5 per week in the treatment group compared with the control group with 90% power and 5% significance, 16 participants for each group were required to complete the trial. To account for an expected 25% dropout, 20 subjects were recruited for each of the two subject populations, or a total of 40 subjects. On examination of the protocol and power calculations, if this study was designed with 80% power for a *p* = 0.05 one-tailed test, it will have approximately 70% power for a *p* = 0.05 two-tailed test. With *n* = 19, it appears that this study would have approximately 79% power for a *p* = 0.05 two-tailed test.

The cytokine data were analysed using analysis of variance (ANOVA) with the factors group (Healthy, IBS) and treatment (Baseline, Kiwifruit flesh, Kiwifruit flesh and skin) and person as a block term. Data for IL-6 and IL-10 were skewed; the variance was stabilised by adding 0.1 and then log transforming the data. The mean number of CSBMs, total bowel movements, CBMs, SBMs, bowel movements with straining and Bristol Stool Score were calculated for the 2 weeks of the lead-in period (baseline) and the last week of the two treatment periods. This number was multiplied by seven to give the measurement per week and the data were analysed using ANOVA, as described above for the cytokine data. The Birmingham IBS score was completed at recruitment and at visits after 2, 4, 6, 8, 10, 12, 14 and 16 weeks. This self-completed questionnaire consists of 14 questions based on the frequency of IBS-related symptoms using a 6-point Likert scale ranging from 1 = all of the time to 6 = none of the time. Following Roalfe et al. [18], the questions were grouped into three domains of constipation, diarrhoea and pain. Baseline was week 2, and weeks 6 and 14 were the treatment data. One participant from the healthy group and 4 participants from the IBS-C group gave no responses to the questionnaire at the beginning of the trial and were excluded from the analysis. The data were analysed using ANOVA as described above and 1 was added to the mean scores and the log values taken to stabilise the variance. Hb1Ac concentrations were compared using a paired t-test.

## 3. Results

### 3.1. Participants

Forty-six individuals were screened and, of these, 40 subjects met the inclusion criteria for this study. Thirty-eight participants completed this study. Two participants withdrew from this study: one withdrawal occurred shortly after the first lead-in period due to health issues that were unrelated to the study products or interventions. One participant withdrew due for personal reasons (Figure 1).

There were no significant differences in the baseline blood biochemistry as analysed by the Chem-20 panel and results were in reference ranges (data not shown). Self-reported compliance with consumption of the study products was 87% and the products were well tolerated, with minor side effects such as flatulence being the only adverse events relating to the study product. The demographics of the participants are shown in Table 2. The values for HbA1c in all participants were in the normal range (<40 mmol/mol) and there were no significant differences between the beginning and the end of this study for the participants in either group.

### 3.2. Primary Outcome

The primary measurement of the trial was change in the concentrations of pro-inflammatory cytokines (TNF-α, IL-6) and an anti-inflammatory cytokine (IL-10) in the blood compared with the baseline. Cytokines were measured at the end of each treatment period and in the washout period and are presented in Table 3 and Figure 2. There was a significant decrease in TNF-α compared with the baseline measurement (*p* < 0.001) when participants from both the healthy and IBS-C groups ate the kiwifruit and skin. While there was a significant treatment effect (*p* < 0.001), there was no group or treatment x group effect. For IL-6, there were no significant differences for treatment, group or treatment x group. Mean concentrations of IL-6 were variable and numerically higher in the blood for both treatments in the healthy group and were lower in the IBS-C group. For IL-10, there were no significant treatment or treatment x group effects but there was a significant difference between the healthy and IBS-C participants. Mean concentrations of IL-10 were higher and more variable in the healthy participants. In the IBS-C group, IL-10 was also significantly lower than baseline following the kiwifruit flesh-only intervention (*p* = 0.003). Further, C-reactive protein (CRP) was measured in the blood of the participants at baseline and at the end of each intervention phase (data not shown). There were no significant effects of either the kiwifruit flesh or kiwifruit flesh and skin treatments on the level of CRP observed in either participant group.

### 3.3. Secondary Outcomes

#### 3.3.1. Stool Frequency and Consistency

Secondary outcomes for this study were changes in bowel movement frequency per week as assessed through the measurement of the number of complete spontaneous bowel movements (CSBMs), total number of bowel movements, complete bowel movements (CBMs), spontaneous bowel movements (SBMs), and the number of strained bowel movements. Additionally, changes in the form and consistency of bowel movements were measured through the daily bowel diaries and comparisons with the Bristol Stool Scale. Table 4 presents the data for bowel movement frequency, Bristol Stool Scale scores, and the proportion of bowel movements requiring laxatives.

There were significant treatment effects for the frequencies of CSBMs, total bowel movements, CBMs, SBMs, strained bowel movements as well as the Bristol Stool Scale. There was a significant effect of participant group for the strained bowel movements frequency (*p* = 0.004) and no treatment x group effects. CSBMs increased in both participant groups after consuming the SunGold kiwifruit and kiwifruit and skin treatments. This was statistically significant for the IBS-C participants on the kiwifruit and skin treatment compared with baseline (*p* < 0.001). Similarly, the total bowel movements per week, CBMs, and SBMs were numerically higher for all the participants consuming the kiwifruit and skin treatment. A significant difference from baseline (*p* < 0.001) was observed in CBMs for IBS-C participants eating kiwifruit and skin (6.7 vs. 9.3). Strained bowel movements decreased for both participant groups eating both treatments (*p* = 0.004 for treatment; *p* = 0.007 for group). Bristol Stool Scale was significantly higher than baseline for the healthy group for both kiwifruit treatments. The proportion of bowel movements using laxatives are presented here but these data were not analysed, as 92% of the participants did not use laxatives. There were no significant interactions of treatment x group for the bowel frequency data.

#### 3.3.2. Gastrointestinal Symptoms (GSRS)

Participants rated their degree of gastrointestinal discomfort using the gastrointestinal symptom rating scale (GSRS) comprising fifteen questions relating to gastrointestinal discomfort rated on a 1–7 scale (no discomfort to very severe discomfort). Following Svelund et al. [19], the GSRS data were grouped into five subscales or domains: diarrhoea, indigestion, constipation, abdominal pain, and reflux (Table 5). There was a significant treatment effect for the constipation and abdominal pain domains, and significant group effects for all five domains. Constipation was significantly reduced for the IBS-C participants for both kiwifruit treatments when compared with the baseline (*p* = 0.004). As expected, the baseline scores for the five domains were significantly higher at baseline for the IBS-C participants than the healthy participants. There was a significant treatment x group effect for the constipation domain (*p* = 0.018).

#### 3.3.3. Gastrointestinal Symptoms Birmingham IBS Questionnaire

The Birmingham IBS gastrointestinal symptom scores are shown in Table 6. There was a significant treatment effect for the constipation and pain scores, and group effect for all three parameters (constipation, diarrhoea, pain). There was a significant reduction in pain scores for the IBS-C participants when they ate the SunGold kiwifruit flesh and skin (*p* = 0.030). The IBS-C participants had significantly higher scores for constipation and diarrhoea than the healthy participants at baseline. For the constipation domain, these scores remained significantly higher than the healthy participants after consuming both kiwifruit treatments. There was no significant treatment x group effect on these scores.

#### 3.3.4. Dietary Intake

Participants were asked to complete a dietary eating habits questionnaire before the start of this study to assess background habits that may relate to health. The parameters measured related to general eating behaviours and practices. Individuals were asked to comment on their current body weight and weight control methods. Most participants (92%) reported that they have a good appetite and consume a varied diet (76%) at regular meal times (Table 7, Figure 3). Only 53% of the participants were happy with their current body weight and would like to reduce their weight by 5 to 10 kg; however, they were overall not concerned about their weight status (67%) and were not actively trying to lose weight (data not shown).

Participant dietary intakes were assessed at the beginning and end of each intervention period. Analysing the dietary intake data without the kiwifruit included in the intake values (Table 8), we found treatment effects for polyunsaturated fatty acids (PUFAs), dietary fibre, potassium, and magnesium. There were no group effects on dietary intake, but there were significant treatment × group effects for saturated fatty acids (SFAs) and magnesium. The results showed that in the healthy group, there was a significant decrease in SFA intake during the kiwifruit and skin treatment compared with the baseline, and a significant decrease in magnesium intake during the kiwifruit and the kiwifruit and skins interventions compared with the baseline (*p* = 0.0055). There were no significant differences in any of the dietary macronutrient or micronutrient intakes of the IBS-C participants for any treatment.

Analysing the dietary intake data including the kiwifruit (Table 9), we found treatment effects for PUFAs, available carbohydrate (CHO), sugar, dietary fibre, and vitamin C (*p* = 0.0055). There were no group effects on dietary intake, but there were significant treatment × group effects for SFAs and magnesium. Compared with the baseline data, the healthy group participants had higher sugar intakes during the kiwifruit flesh treatment and higher vitamin C intakes during both kiwifruit interventions. The IBS-C participants consumed more available CHO, sugar and vitamin C during the kiwifruit flesh treatment, and higher intakes of sugar and vitamin C during the kiwifruit and skin treatment. The healthy participants consumed more SFAs during the kiwifruit and skins intervention than that measured at baseline.

## 4. Discussion

The primary outcome of the trial was to examine changes in the concentrations of pro-inflammatory cytokines (TNF-α, IL-6) and anti-inflammatory cytokine (IL-10) in the blood following consumption of Zespri^®^ SunGold kiwifruit with and without skin. This was measured in a population of 19 healthy individuals and a population of 19 individuals with IBS-C. We found that the consumption of kiwifruit reduced the concentrations of the pro-inflammatory cytokine, TNF-α, but had no effect on IL-6 and IL-10 concentrations. We also observed that the healthy participants had higher concentrations of IL-10 than the IBS-C participants and that overall inflammation was not affected by kiwifruit consumption as measured by C-reactive protein.

TNF-α is a pro-inflammatory cell signalling protein (cytokine) that plays a key role in the regulation of immune cells [20]. TNF-α can initiate both acute and chronic inflammation and increases production of the cytokine IL-6. IL-6 is involved in inflammation and infection responses as well as the regulation of metabolic, regenerative and neural processes. Whilst, in many circumstances, TNF-α and IL-6 are critical for immune defence, they are also associated with uncontrolled systemic inflammation. This can be characterized by an underlying pro-inflammatory state, distinguished by increased circulating concentrations of C-reactive protein (CRP) and inflammatory cytokines (e.g., IL-6 and TNF-α) as well as decreased concentrations of anti-inflammatory mediators such as adiponectin and IL-10 [21,22,23]. IL-10 is an anti-inflammatory cytokine that plays an important role in regulating the concentrations of other cytokines, particularly the synthesis of pro-inflammatory cytokines such as TNF-α [24]. There is growing evidence that inflammation plays an important role in IBS and recent studies have explored cytokines as biomarkers for the condition [25]. IL-6 has been shown to be elevated in patients with IBS [26], whilst IL-10 was found to be lower in individuals with IBS [24]. In the present study, we observed a decreased concentration of IL-10 in individuals with IBS-C compared with the healthy control individuals when they consumed kiwifruit flesh. Furthermore, another study showed that peripheral blood mononuclear cells from IBS patients had higher baseline TNF-α and IL-6 than healthy controls [27]. However, when the IBS group was split into subgroups of patients with IBS-C and IBS with diarrhoea (IBS-D), patients with IBS-D had the highest TNF-α and IL-6. We did not observe any differences in concentrations of TNF-α and IL-6 at baseline between the IBS-C and healthy individuals due to our selection criteria being focused on IBS-C as the predominant condition.

Components of gold kiwifruit, such as vitamin C, vitamin B9-folate, vitamin A, lutein and zeaxanthin, have also been implicated in supporting immune function [28] largely through their anti-oxidant properties. In addition, the presence of the unique proteins actinidin and kiwellin is thought to contribute to the potential health benefits of kiwifruit and their role in supporting immunity [29]. While there is some evidence to suggest that kiwifruit consumption may modulate cytokine regulation, there have been no reports of definitive changes in the concentrations of pro and anti-inflammatory cytokines.

TNF-α and IL6 were not significantly different between the healthy and IBS-C participants at baseline. These are not well-established biomarkers and there are no published references that the authors are aware of that show a reference range for these cytokines in healthy individuals. The literature will state however that IL-6 and TNF-α are often significantly higher in people with a range of metabolic disorders. The baseline screening process of participants was limited to a Chem 20 panel and therefore may not have accounted for all underlying inflammatory conditions that could have caused variation leading to some confounding. This is a limitation of this study and more detailed screening of individuals at enrollment might explain this.

The secondary outcomes of this study investigated the potential of consuming SunGold kiwifruit to relieve gastrointestinal discomfort and improve bowel motion frequency in individuals with IBS-C in comparison to healthy individuals. Complete spontaneous bowel movements (CSBMs) increased by more than 1 per week in the healthy participants when the skin was also consumed, and increased by more than 1 per week when the IBS-C participants consumed the kiwifruit flesh only and by more than 2 per week when consuming the flesh and the skin. An increase of greater than one complete spontaneous bowel motion per week is considered a clinically significant marker of improved bowel function in individuals symptomatic of constipation [30]. In addition, increases were observed in total bowel movement, complete bowel movement, and spontaneous bowel movement frequencies, which were greater with the kiwifruit and skin treatment than the kiwifruit flesh alone. Straining during bowel movements was also reduced significantly and stool form was improved.

Normal bowel movement frequency and consistency are highly variable among individuals and populations due to several different factors including dietary habit, gut transit time and fibre intake. What is considered normal bowel movement ranges from three stools per day to three per week and some level of straining and incomplete evacuation is usual [31]. In this group of study participants, the Rome III criteria were used to correctly classify the IBS-C group but this questionnaire also highlighted that the healthy group of participants had wide variations in the number and consistency of bowel movements that could be improved by the addition of kiwifruit to their diets. 

The increase in CSBM and CBM frequencies in the IBS-C group with kiwifruit flesh and skin provides a dietary means to improve their daily well-being. On the basis of prior research [29], the effect of kiwifruit flesh alone and kiwifruit and skin can largely be attributed to the addition of dietary fibre to the habitual diet of the participants. Three kiwifruit per day equated to 360 g fruit and provided participants with 5 g of fibre per day [29]. Addition of the kiwifruit skin would have added yet more insoluble fibre. Dietary fibre is a vital part of the diet that is generally accepted to be an essential part of maintaining gut health [32]. The presence of both soluble and insoluble fibre in kiwifruit is thought to contribute to the demonstrated beneficial effect on laxation.

The dietary intake data show that the participants in this study generally had a dietary fibre intake lower than the recommended daily intake, which is set at 25 g/day for women and 30 g/day for men [33]. Baseline data show that the mean intake was 23 g/day before the trial treatment. It appears that participants may then have adjusted their intake of fibre-rich foods further to compensate for the addition of 3 kiwifruit/day as fibre intake assessed without the kiwifruit contribution was at an average of 18.5 g/day for the healthy group and 21.5 g/day for the IBS-C group. With addition of the kiwifruit to the dietary analysis, the figures for the healthy group remained at 23 g/day for the healthy group but increased to 26 g/day for the IBS-C group, which may explain the greater increase in CSBMs and CBMs for this group. Increasing dietary fibre intake is a commonly recommended treatment for the relief of IBS-C symptoms in clinical practice [34] and the results of this study suggest that kiwifruit flesh alone and kiwifruit flesh and skin can be an effective dietary treatment for people with IBS-C.

Gastrointestinal discomfort arising from constipation manifests in several different symptoms, which can considerably affect an individual’s health [35]. Using the gastrointestinal symptom rating scale (GSRS), this study also examined the level of gastrointestinal comfort across five domains of diarrhoea, indigestion, constipation, abdominal pain, and reflux. In the healthy group, there were no significant differences observed in any of the domains; however, there was a significant effect in the constipation domain for the IBS-C group. The Birmingham IBS questionnaire was also included as a further subjective measure of gastrointestinal comfort measuring three domains of constipation, diarrhoea and pain. Whole kiwifruit consumption (flesh and skin) significantly reduced the pain score for the IBS-C participants compared with the baseline.

Dietary intake was also assessed at the end of each intervention period to monitor any changes to dietary intake over the course of this study. Data were analysed both with and without the contribution made by the trial product. There was no significant effect on total energy intake during the course of this study for either set of data. Overall, the participants reported lower total energy intakes with an average daily intake of approximately 7500 kJ total energy as opposed to the adult recommendations of 8000 kJ for women and 10,000 kJ for males [36]. Including the nutrient contribution from the kiwifruit resulted in a significant increase in sugar intake, vitamin C content, fibre and available carbohydrate content across both groups when compared with the lead-in period. This trial ran over three seasons: autumn, winter and spring, which naturally bring around seasonal changes in diet that would account for small variations in intake. The diet records obtained during this study provide only a snapshot of food habits and these self-reporting tools are challenging in their degree of reliability. People do not always pay attention to the food they are eating, do not remember everything, do not know the ingredients of the foods they are eating, and underestimate portion size. Factors such as gender and weight status can also bias recording [37]. The true effect of the intervention inclusion on individual daily dietary intake would require a trial that had a greater emphasis on this area, but our results suggest that including the trial products into the daily diet did not influence the participant’s nutrient intakes.

The palatability and tolerability of the study products were rated as high by participants and no adverse effects relating to either of the study products were reported. Compliance was high, with 87% of participants reporting that they took the interventions as requested. This may be because kiwifruit is already widely used by constipated individuals in the general population and is anecdotally considered to be one of the most preferred interventions. As such, many of the participants were already aware that they could tolerate SunGold kiwifruit, which was highly appealing to the participants and well liked A further limitation of this study is that consumption of the skin of kiwifruit is not appealing to all individuals and compliance with this aspect may have been difficult to adhere to, although we did not have any reports of this in the trial.

## 5. Conclusions

This randomized, controlled study examined the consumption of three Zespri^®^ SunGold kiwifruit daily in healthy individuals and individuals with IBS-C and aimed to show that consuming whole fruit (with and without the skin) of SunGold kiwifruit would affect the production of the inflammatory marker C-reactive protein and cytokines IL-6, IL-10 and TNF-α in healthy individuals and those with IBS-C. We found the consumption of kiwifruit reduced the concentrations of the pro-inflammatory cytokine, TNF-α, but had no effect on IL-6, IL-10 or C-reactive protein concentrations in the blood. Clinically significant changes were observed in the healthy and IBS-C participants of greater than 1 CSBM per week and was enhanced in the IBS-C study population, in which CSBMs increased to greater than 2 per week with the kiwifruit flesh and skin. Symptoms of gastrointestinal discomfort were also improved in the IBS-C participants, with a decrease in constipation.

This study is the first to assess the impact of consuming the skin with the flesh for SunGold kiwifruit. We have demonstrated that consuming the skin of kiwifruit might have beneficial effects on gastrointestinal health that are not produced by consuming the flesh alone. Further studies in this area are required to expand on these findings using the same and other populations. This will ascertain whether there are similar effects in other populations and to investigate the mechanisms of action that drive these effects.

## Figures and Tables

**Figure 1 nutrients-12-01453-f001:**
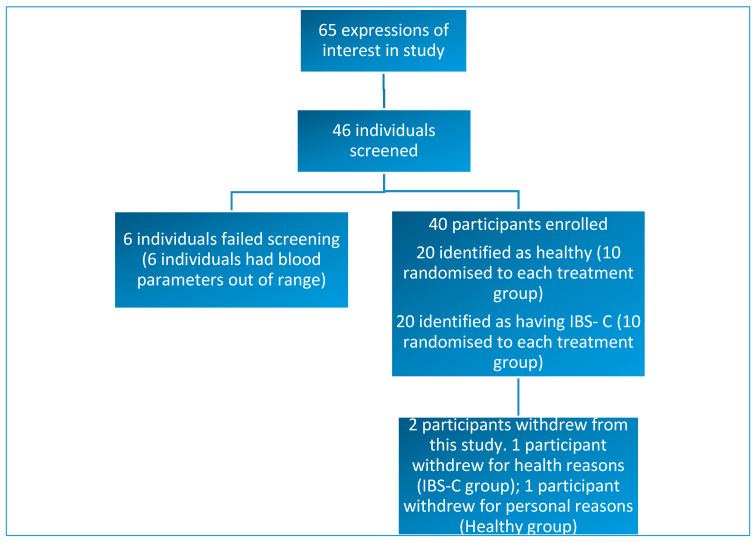
Diagram showing study enrolment and reasons for exclusion or withdrawal from this study.

**Figure 2 nutrients-12-01453-f002:**
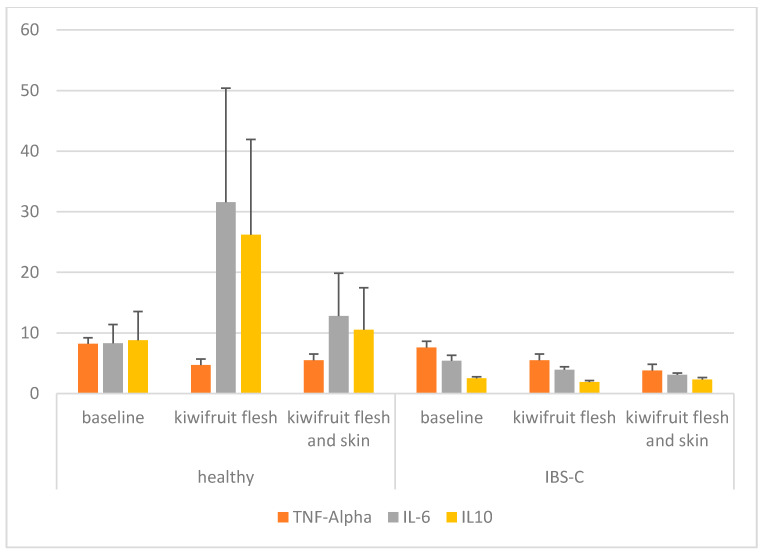
Bar graph of blood cytokine concentrations (pg/mL) of the participants consuming SunGold kiwifruit with and without skin, grouped into healthy individuals and those with IBS-C.

**Figure 3 nutrients-12-01453-f003:**
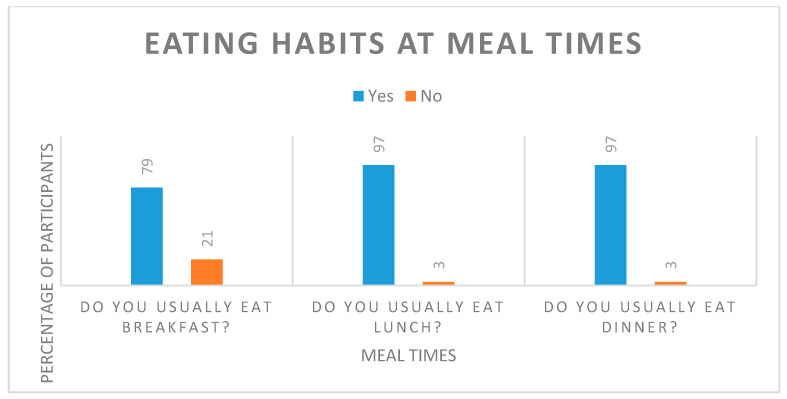
Eating habits at standard meal times reported by study participants.

**Table 1 nutrients-12-01453-t001:** Irritable bowel syndrome (IBS) diagnostic criteria *.

Recurrent Abdominal Pain on Average at Least 1 Day/Week in the Last 3 Months, Associated with Two or More of the Following Criteria:
1. Related to defecation
2. Associated with a change in the frequency of stool
3. Associated with a change in the form (appearance) of stool

(These criteria should be fulfilled for the last 3 months, with symptom onset at least 6 months prior to diagnosis.) * Modified from Rome IV.

**Table 2 nutrients-12-01453-t002:** Demographics and baseline characteristics of participants in this study.

Baseline Characteristics	Total Number
Number of study participants	38
Male	11
Female	27
Ethnicity	
New Zealand European	32
Māori	1
Israeli	1
South American	2
Russian	2
Age (years) ^1^	44 (22–65)
Weight (kg) ^1^	71 (52–109)
BMI ^1^	25 (19–34)

^1^ Data presented as median (range); BMI = Body Mass Index.

**Table 3 nutrients-12-01453-t003:** Blood cytokine concentrations (pg/mL) of the participants consuming SunGold kiwifruit with and without skin ^1^.

Participant Group	Trial Phase	TNF-α	IL-6	IL-10
Healthy	Baseline	8.2 (3.0)	8.3 (13.4)	8.8 (20.7)
	Kiwifruit flesh	4.7 * (1.6)	31.6 (82.0)	26.2 (68.5)
	Kiwifruit flesh and skin	5.5 * (3.8)	12.8 (30.8)	10.5 (30.4)
IBS-C	Baseline	7.6 (2.9)	5.4 (3.9)	2.5 (1.1)
	Kiwifruit flesh	5.5 (4.0)	3.9 (2.3)	1.9 ^ (1.1)
	Kiwifruit flesh and skin	3.8 * (1.5)	3.1 (1.2)	2.3 (1.5)
*p* Values				
Treatment		<0.001	0.141	0.083
Group		0.476	0.065	0.302
Treatment × Group		0.083	0.003	0.370

^1^ Data presented as the mean (standard deviation); * mean significantly different from baseline based on least significant difference (*p* = 0.0055); ^ mean significantly different from same treatment in healthy group based on least significant difference (*p* = 0.0055).

**Table 4 nutrients-12-01453-t004:** Bowel movement (BM) frequency and Bristol Stool Scale results for participants consuming Zespri^®^ SunGold kiwifruit (*Actinidia chinensis* var. *chinensis* ‘Zesy002’) with and without skin ^1^.

Participant Group	Trial Phase	No. of CSBMs Per Week	No. of BMs Per Week	No. of CBMs Per Week	No. of SBMs Per Week	No. of Strained BMs Per Week	Bristol Stool Scale
Healthy	Baseline	7.6 (5.4)	10.5 (5.2)	9.7 (5.6)	8.3 (5.2)	1.1 (1.5)	3.3 (0.7)
	Kiwifruit flesh	7.8 (5.1)	10.1 (5.3)	9.7 (5.6)	8.1 (5.0)	0.6 (1.0)	4.0 * (1.0)
	Kiwifruit flesh and skin	9.8 (5.7)	11.4 (5.7)	11.1 (6.0)	10.1 (5.4)	0.1 (0.3)	3.9 * (0.7)
IBS-C	Baseline	6.7 (4.8)	9.5 (5.6)	6.7 (4.8)	9.4 (5.6)	2.8 (2.9)	3.5 (1.0)
	Kiwifruit flesh	8.4 (7.0)	11.0 (7.5)	8.7 (6.9)	10.7 (7.7)	2.3 (2.7)	3.9 (1.0)
	Kiwifruit flesh and skin	9.2 * (5.8)	11.4 (6.2)	8.7 (6.9)	11.3 (6.1)	1.9 (2.5)	3.8 (1.1)
*p* Values							
Treatment		<0.001	0.020	<0.001	0.009	0.007	<0.001
Group		0.871	0.995	0.303	0.377	0.004	0.953
Treatment × Group		0.405	0.174	0.151	0.422	0.972	0.390

^1^ Data presented as the mean (standard deviation); complete spontaneous bowel movement (CSBM); complete bowel movement (CBM); spontaneous bowel movement (SBM); * mean significantly different from baseline based on least significant difference (*p* = 0.0055).

**Table 5 nutrients-12-01453-t005:** Gastrointestinal symptom rating scale (GSRS) for the participants consuming Zespri^®^ SunGold kiwifruit (*Actinidia chinensis* var. *chinensis* ‘Zesy002’) with and without skin ^1^.

Participant Group	Trial Phase	Diarrhoea	Indigestion	Constipation	Abdominal Pain	Reflux
Healthy	Baseline	1.10 (0.13)	1.37 (0.34)	1.29 (0.35)	1.25 (0.24)	1.13 (0.23)
	Kiwifruit flesh	1.04 (0.11)	1.36 (0.41)	1.21 (0.37)	1.11 (0.19)	1.13 (0.23)
	Kiwifruit flesh and skin	1.21 (0.47)	1.43 (0.49)	1.05 (0.12)	1.28 (0.39)	1.16 (0.34)
IBS-C	Baseline	1.94 ^ (1.09)	2.48 ^ (1.11)	2.67 ^ (1.11)	2.05 ^ (0.89)	1.55 (0.91)
	Kiwifruit flesh	2.09 ^ (1.55)	2.10 (1.26)	1.73 * (0.95)	1.67 (0.95)	1.34 (0.57)
	Kiwifruit flesh and skin	2.15 ^ (1.59)	2.44 ^ (1.28)	2.04 * ^ (1.15)	2.02 ^ (1.12)	1.56 (0.91)
*p* Values						
Treatment		0.342	0.149	0.004	0.036	0.584
Group		<0.001	0.001	<0.001	0.002	0.023
Treatment × Group		0.788	0.241	0.018	0.528	0.623

^1^ Data presented as the mean (standard deviation); * mean significantly different from baseline based on least significant difference (*P* = 0.0055); ^ mean significantly different from same treatment in healthy group based on least significant difference (*p* = 0.0055).

**Table 6 nutrients-12-01453-t006:** Gastrointestinal discomfort measured by Birmingham IBS questionnaire for the participants consuming Zespri^®^ SunGold kiwifruit (*Actinidia chinensis* var. *chinensis* ‘Zesy002’) with and without skin ^1^.

Participant Group	Trial Phase	Constipation	Diarrhoea	Pain
Healthy	Baseline	0.44 (0.40)	0.19 (0.20)	0.37 (0.32)
	Kiwifruit flesh	0.35 (0.46)	0.22 (0.33)	0.30 (0.41)
	Kiwifruit flesh and skin	0.22 (0.34)	0.23 (0.27)	0.22 (0.30)
IBS-C	Baseline	1.82 ^ (1.37)	0.84 ^ (0.73)	0.89 (0.81)
	Kiwifruit flesh	1.62 ^ (0.99)	0.61 (0.53)	0.63 (0.64)
	Kiwifruit flesh and skin	1.22 ^ (1.16)	0.51 (0.47)	0.42 * (0.54)
*p* Values				
Treatment		0.014	0.376	0.004
Group		<0.001	0.001	0.030
Treatment × Group		0.626	0.105	0.361

^1^ Data presented as the mean (standard deviation); * mean significantly different from baseline based on least significant difference (*p* = 0.0055); ^ mean significantly different from same treatment in Healthy group based on least significant difference (*p* = 0.0055).

**Table 7 nutrients-12-01453-t007:** General eating habits of the study participants.

Question	Response	N ^1^ (%)
How would you describe your appetite?	Good Fair Poor	35 (92) 2 (5.5) 1 (2.6)
How diverse is your diet?	Different every day Different only sometimes during the week Different only during weekend days Very monotonous	16 (42) 13 (34.2) 3 (7.9) 6 (15.8)

^1^*n* = 38. Good—able to eat and enjoy moderate-sized meals easily every day and snack between meals. Moderate—able to eat moderate-sized meals but not always complete meals and seldom snacking between meals. Poor—never feeling like eating or being hungry and not enjoying eating.

**Table 8 nutrients-12-01453-t008:** Daily dietary intakes of selected nutrients by the participants excluding the Zespri^®^ SunGold kiwifruit (*Actinidia chinensis* var. *chinensis* ‘Zesy002’) interventions ^1^.

Nutrient	Baseline		Kiwifruit Flesh Only		Kiwifruit and Skin		*p* Value		
	Healthy	IBS	Healthy	IBS	Healthy	IBS	Treatment	Group	Treatment × Group
Total energy (kJ)	8083 (2693)	7258 (1858)	7159 (2541)	7877 (2885)	6963 (2211)	6829 (2255)	0.078	0.911	0.105
Protein (g)	92 (50)	72 (25)	82 (30)	75 (29)	80 (19)	65 (25)	0.333	0.090	0.530
Total fat (g)	81 (35)	73 (23)	70 (29)	71 (32)	67 (24)	69 (35)	0.062	0.853	0.396
Saturated fatty acids (g)	32 (12)	25 (10)	27 (10)	28 (12)	24 * (11)	28 (15)	0.362	0.819	0.007
Polyunsaturated fatty acids (g)	12 (11)	13 (6)	10 (8)	13 (7)	10 (6)	9 (6)	0.013	0.662	0.122
Monounsaturated fatty acids (g)	28 (13)	28 (11)	23 (11)	28 (14)	25 (9)	24 (15)	0.091	0.806	0.187
Available carbohydrate (g)	188 (59)	172 (56)	170 (67)	193 (106)	169 (78)	161 (65)	0.269	0.979	0.192
Sugar (g)	76 (33)	77 (23)	67 (44)	75 (32)	63 (38)	70 (29)	0.080	0.622	0.667
Dietary fibre (g)	23 (16)	23 (5)	20 (13)	24 (9)	17 (7)	19 (7)	0.004	0.486	0.427
Vitamin C (mg)	67 (79)	80 (28)	63 (51)	74 (41)	97 (79)	73 (47)	0.474	0.991	0.329
Sodium (mg)	2596 (1076)	2163 (1020)	2249 (962)	2038 (878)	2029 (1117)	1933 (689)	0.139	0.327	0.674
Potassium (mg)	3032 (1300)	3159 (632)	2373 (977)	3074 (768)	2546 (744)	2642 (918)	0.005	0.235	0.096
Magnesium (mg)	332 (160)	328 (106)	247 * (123)	349 (140)	256 * (95)	292 (144)	0.002	0.277	0.004
Iron (mg)	12 (6)	12 (4)	10 (4)	12 (5)	11 (4)	10 (4)	0.188	0.744	0.157
Calcium (mg)	752 (352)	798 (382)	700 (298)	781 (326)	651 (291)	701 (365)	0.202	0.534	0.939

^1^ Data presented as the mean (standard deviation); * mean significantly different from baseline based on least significant difference (*p* = 0.0055).

**Table 9 nutrients-12-01453-t009:** Daily dietary intakes of selected nutrients by the participants including the Zespri^®^ SunGold kiwifruit (*Actinidia chinensis* var. *chinensis* ‘Zesy002’) interventions ^1^.

Nutrient	Baseline		Kiwifruit Flesh Only		Kiwifruit and Skin		*p* Value		
	Healthy	IBS	Healthy	IBS	Healthy	IBS	Treatment	Group	Treatment × Group
Total energy (kJ)	8083 (2693)	7258 (1858)	7758 (2551)	8444 (2873)	7543 (2229)	7432 (2256)	0.237	0.907	0.116
Protein (g)	92 (50)	72 (25)	84 (31)	78 (29)	83 (19)	68 (25)	0.505	0.090	0.539
Total fat (g)	81 (35)	73 (23)	71 (29)	78 (29)	68 (24)	70 (35)	0.073	0.975	0.132
Saturated fatty acids (g)	32 (12)	25 (10)	27 (10)	28 (12)	24 * (11)	28 (15)	0.405	0.822	0.007
Polyunsaturated fatty acids (g)	12 (11)	13 (6)	10 (8)	13 (7)	10 (6)	10 (6)	0.026	0.670	0.124
Monounsaturated fatty acids (g)	28 (13)	28 (11)	23 (11)	28 (14)	25 (9)	24 (14)	0.098	0.807	0.180
Available carbohydrate (g)	188 (59)	172 (56)	200 (67)	221 * (106)	198 (78)	190 (65)	0.036	0.969	0.233
Sugar (g)	76 (33)	77 (23)	96 * (44)	103 * (31)	91 (40)	99 * (29)	<0.001	0.650	0.715
Dietary fibre (g)	23 (16)	23 (5)	24 (13)	28 (9)	21 (7)	23 (7)	0.042	0.484	0.458
Vitamin C (mg)	67 (79)	80 (28)	446 * (61)	429 * (92)	471 * (104)	454 * (54)	<0.001	0.643	0.604
Sodium (mg)	2596 (1076)	2163 (1020)	2260 (960)	2089 (878)	2035 (1116)	1940 (689)	0.148	0.324	0.677
Potassium (mg)	3032 (1300)	3159 (632)	3117 (989)	3787 (778)	3276 (755)	3402 (916)	0.079	0.237	0.140
Magnesium (mg)	332 (160)	328 (106)	277 (123)	377 (140)	285 (95)	322 (144)	0.181	0.278	0.005
Iron (mg)	12 (6)	12 (4)	10 (4)	13 (5)	11 (4)	11 (4)	0.447	0.739	0.164
Calcium (mg)	752 (352)	798 (382)	741 (299)	822 (327)	691 (293)	743 (365)	0.454	0.532	0.945

^1^ Data presented as the mean (standard deviation); * mean significantly different from baseline based on least significant difference (*p* = 0.0055).

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
