# Peer review of "The Effects on Immune Function and Digestive Health of Consuming the Skin and Flesh of Zespri® SunGold Kiwifruit (Actinidia Chinensis var. Chinensis ‘Zesy002’) in Healthy and IBS-Constipated Individuals"

_nutrients, 2020, doi:10.3390/nu12051453_

Round 1
Reviewer 1 Report
1.- It would be convenient to include in the title after the kiwifruit on "the efficacy on healthy and IBS-constipated people"
2.- The Introduction section is a litle long. You might shorten some phrases or paragraphs and move them to the Discussion Section
3.- The Material and Methods Section is well described and with very clear content
4.- The Results Section is very complete and the Tables are very clearly described and very easy its reading
5.- The Discussion is good interpretan very well the results obtained
6.- The conclusions are clear and pertinent
Author Response
Reviewer 1
- It would be convenient to include in the title after the kiwifruit on “the efficacy on healthy and IBS-constipated people”
We agree with your suggestion. The title is now (line 5)
“The effects on immune function and digestive health of consuming the skin and flesh of Zespri® SunGold Kiwifruit (Actinidia chinensis var. chinensis ‘Zesy002’) in healthy and IBS-constipated individuals”
- The introduction is a little long. You might shorten some phrases or paragraphs and move them to the Discussion section.
We acknowledge that our introduction is long but feel it is necessary in order to explain to the reader why we did the study and that there is a sound basis for the study.
Reviewer 2 Report
Eady et al. conducted a study investigating the effect of consuming Zespri@ SunGold Kiwifruit with or without skin on cytokine levels and gut health in patients with IBS-C. The effect of kiwifruit consumption on cytokines is minimal, however, participants had kiwifruit flesh and skin display improved bowel movements and gastrointestinal discomfort. This study is of interest, and I have only minor comments.
- Authors should provide more details on explaining CSBM, CBM, SBM, strained bowel movements and all. What are the difference among those parameters?
- In figure 1, the diagram should also include how the participants were assigned to kiwi flesh group or flesh+skin group.
- There is no figure legend for figure 2. Could author separate the healthy participants and IBS-C in this figure?
- Why the baseline of TNF-a and IL-6 in IBS-C participants is lower than those in healthy participants? Author could discuss on this a little further.
- Please include all the questionnaire forms as supplementary files, which will provide useful information for other readers.
Reviewer 3 Report
I read with interest the MS "The effects on immune function and digestive health 2 of consuming the skin and flesh of Zespri® SunGold 3 Kiwifruit (Actinidia chinensis var. chinensis 4 ‘Zesy002’)" by Eady SL and coworkers. IBS patients are constantly looking for alternative treatments and any effort to provide sound data is welcomed. However, I found major biases, namely: a) inclusion criteria are unclear the Author mention "mild" IBS-C as an inclusion criteria, but there is not such a score in the Rome III criteria, in addition a number of patients seem to complain of diarrhea thus qualifying for mixed bowel pattern, b) recruitment procedure is unclear, I would expect two tails recruitment for controls and patients, c) Obese subjects were apparently excluded, but Authors failed to address the issue, d) Authors claim their data support the assumption of kiwifruit with flesh. However it looks to me that the difference between Kiwifruit with and without flesh is not significant and this needs to be properly addressed, e) a would suggest to add ethnicity as a potential limitation if assuming kiwifruit with flesh is eventually suggested, I doubt Europeans would eat kiwifruit with flesh particularly at this new pandemic era. Minor suggestions: a) the Authors quote general digestive health a number of times, such a phrasing is to be avoided for being not specific, b) ref #13 is an abstract and should be dropped for there are published papers addressing the issue, c) I would suggest to include a Rome III IBS diagnostic table for sake of clarity, d) consideration of adding a bar table for primary aim would be much welcomed, e) BMI reported in the paper is at variance with the one reported in the dedicated table
Author Response
See attached word file

Round 2
Reviewer 3 Report
I read with interest the revised version of the MS "The effects on immune function and digestive health 2 of consuming the skin and flesh of Zespri® SunGold 3 Kiwifruit (Actinidia chinensis var. chinensis 4 ‘Zesy002’) in healthy and IBS-constipated individuals" by Eady SL and coworkers. The Authors addressed partially my queries. Regrettably, I believe explaining to the reviewer the diagnostic choice of "mild scoring" of IBS would be better choice than simply dropping it without further justification. Having stated that, the statistic has been improved. However, reported data arguably support the abstract statement "We have demonstrated that consuming the skin of SunGold kiwifruit may have beneficial effects on gastrointestinal health that are not produced by consuming the flesh alone" which needs to be changed toward "We have demonstrated that consuming the skin of SunGold kiwifruit MIGHT have beneficial effects on gastrointestinal health that are not produced by consuming the flesh alone". This modification needs to apply to all of the phrasing of the actual MS.
Author Response
- Regrettably, I believe explaining to the reviewer the diagnostic choice of "mild scoring" of IBS would be better choice than simply dropping it without further justification.
We chose to include only individuals with mild IBS and exclude anyone that had chronic and severe IBS. Those individuals who had extreme pain and constipation symptoms that had persisted for long periods of time requiring frequent laxative and pain medication were excluded. We believe that it would have been unethical to ask these individuals in pain and discomfort to stop all medication they were taking for their IBS for a long time period of 16 weeks. It would also have increased the likelihood of protocol non-compliance with this requirement
2. However, reported data arguably support the abstract statement "We have demonstrated that consuming the skin of SunGold kiwifruit may have beneficial effects on gastrointestinal health that are not produced by consuming the flesh alone" which needs to be changed toward "We have demonstrated that consuming the skin of SunGold kiwifruit MIGHT have beneficial effects on gastrointestinal health that are not produced by consuming the flesh alone". This modification needs to apply to all of the phrasing of the actual MS.
The authors have altered the ‘may’ to ‘might’ in the manuscript.